# An excitatory cortical feedback loop gates retinal wave transmission in rodent thalamus

Yasunobu Murata[1,2], Matthew T Colonnese[1,2]*

[1]Department of Pharmacology and Physiology, George Washington University, Washington, United States; [2]Institute for Neuroscience, George Washington University, Washington, United States

**Abstract** Spontaneous retinal waves are critical for the development of receptive fields in visual thalamus (LGN) and cortex (VC). Despite a detailed understanding of the circuit specializations in retina that generate waves, whether central circuit specializations also exist to control their propagation through visual pathways of the brain is unknown. Here we identify a developmentally transient, corticothalamic amplification of retinal drive to thalamus as a mechanism for retinal wave transmission in the infant rat brain. During the period of retinal waves, corticothalamic connections excite LGN, rather than driving feedforward inhibition as observed in the adult. This creates an excitatory feedback loop that gates retinal wave transmission through the LGN. This cortical multiplication of retinal wave input ends just prior to eye-opening, as cortex begins to inhibit LGN. Our results show that the early retino-thalamo-cortical circuit uses developmentally specialized feedback amplification to ensure powerful, high-fidelity transmission of retinal activity despite immature connectivity.

*For correspondence:
colonnese@gwu.edu

**Competing interests:** The authors declare that no competing interests exist.

## Introduction

Throughout the central nervous system, before the development of sensory input or behavioral experience, specialized circuitry generates spontaneous activity that is required for circuit formation (*Kirkby et al., 2013*). In sensory isocortex the vast majority of spontaneous activity is not generated locally, but rather in the peripheral sense organ (*Khazipov et al., 2013a*). For example, prior to vision the retina generates waves of spontaneous firing that refine receptive field characteristics in visual thalamus and cortex (*Huberman et al., 2008*). Retinal waves drive robust firing in LGN, superior colliculus, and VC (*Ackman and Crair, 2014*) through a complex interaction between these structures (*Weliky and Katz, 1999*) that remains poorly understood. In the retina, transient circuit properties are responsible for wave propagation, most prominently hyper-connectivity of starburst amacrine cells (*Blankenship and Feller, 2010*), but the degree to which the central transmission of retinal waves, or spontaneous activity in other systems, requires similar specializations is unknown.

During the period of retinal waves, early ganglion cell activity undergoes both amplification and transformation in the brain in ways substantially different from the adult transmission of retinal activity, suggesting the presence of unique developmental circuitry. For example, the relative amplification of ganglion cell spiking in developing thalamocortical circuits is approximately 40-fold that of the adult (*Colonnese et al., 2010*). Evidence of similar amplification is observed in late second and early third trimester human neonates (born preterm), who show sensory-evoked and spontaneous EEG events that are paradoxically large, reaching amplitudes of over 1 millivolt, compared to microvolts in adults (*Vanhatalo and Kaila, 2006*). These strong neonatal activities persist despite low synaptic density (5–10% of the adult) (*Aggelopoulos et al., 1989*; *Bourgeois and Rakic, 1993*), weak

**eLife digest** The brain of a developing fetus has a big job to do: it needs to create the important connections between neurons that the individual will need later in life. This is a challenge because the first connections that form between neurons are sparse, weak and unreliable. They would not be expected to be able to transmit signals in a robust or effective way, and yet they do. How the nervous system solves this problem is an important question, because many neurological disorders may be the result of bad wiring between neurons in the fetal brain.

When an adult human or other mammal "sees" an object, visual information from the eye is transmitted to a part of the brain called the thalamus. From there it is sent on to another part of the brain called the cortex. The cortex also provides feedback to the thalamus to adjust the system and often acts as a brake in adults to limit the flow of information from the eyes.

Murata and Colonnese investigated whether the fetal brain contains any "booster" circuits of neurons that can amplify weak signals from other neurons to help ensure that information is transferred accurately. The experiments monitored and altered visual activity in the brains of newborn rats – which have similar activity patterns to those observed in human babies born prematurely. Murata and Colonnese found that in these rats the feedback signals from the cortex to the thalamus actually multiply the visual signals from the eye, instead of restraining them. This causes a massive amplification in activity in the developing brain and explains how the fetal brain stays active despite its neurons being only weakly connected.

The booster circuit stops working just before the eyes first open (equivalent to birth in humans) as the connections between neurons become stronger, and is replaced by the braking mechanism seen in adults. This is important, because continued amplification of signals in the adult brain might cause excessive brain activity and epilepsy. The findings of Murata and Colonnese may therefore help to explain why epileptic seizures have different causes and behave differently in children and adults.

The next step following on from this work is to find out how the braking mechanism forms in young animals. Future studies will also focus on understanding the precise role the booster circuit plays in early brain development.

individual synapses (*Chen and Regehr, 2000*; *Etherington and Williams, 2011*), and poor action-potential reliability (*McCormick and Prince, 1987*). Not only is retinal wave activity amplified in the thalamocortical pathway, it is also transformed into synchronized rapid oscillations that enhance propagation and synaptic plasticity (*Khazipov et al., 2013b*; *Luhmann et al., 2016*). Cortical activity during the period of retinal waves follows nearly exactly the macro-patterning of retinal activity: periods of silence lasting 30–60 s are interrupted by 500 ms–2 s waves that occur in clusters lasting up to 10 s (*Hanganu et al., 2006*; *Colonnese and Khazipov, 2010*; *Ackman et al., 2012*). However, the cortical activity within waves is organized in rapid 'spindle-burst' oscillations (8–30 Hz) (*Hanganu et al., 2006*; *Kummer et al., 2016*) that are not present in retina and not observed in adult cortical activity, suggesting transitory early circuit properties are responsible for their production (*Tolner et al., 2012*). Spindle-bursts are not limited to VC, but are ubiquitous throughout neonatal cortex. They are the rodent homologue of delta-brushes (*Khazipov et al., 2013a*), one of the major components of the human preterm EEG (*André et al., 2010*), and their absence in patients indicates a poor prognosis. Thus understanding their central generative mechanisms would be a critical contribution to neonatal health and development.

We hypothesized that spindle-burst oscillations and retinal-wave amplification arise in LGN, but depend critically on corticothalamic feedback (*Weliky and Katz, 1999*). To test this we recorded simultaneously from LGN and primary VC of head-fixed, unanesthetized rats both during the period of retinal waves (P5-11), and immediately after (P13-14), when mature cortical activity patterns have begun to emerge (*Rochefort et al., 2009*; *Colonnese, 2014*; *Hoy and Niell, 2015*). We investigated the relative roles of retina, thalamus and cortex by pharmacological or optogenetic modulation of each region. Our results demonstrate a developmentally unique role for the corticothalamic

projection that results from the late development of inhibition relative to excitation, and is necessary for the propagation and synchronization of retinal wave-driven activity through thalamus and cortex.

## Results

To determine the thalamic contribution to the generation of spindle-burst oscillations evoked by spontaneous retinal waves, we made simultaneous recordings from the LGN and monocular VC in head-fixed, unanesthetized rats. We focused on post-natal days (P) 9–11, the time of peak stage-III (glutamatergic) retinal wave activity when cortical activity is dominated by 20–30 Hz spindle oscillations that synchronize all layers (*Colonnese and Khazipov, 2010*). Linear multi-electrode arrays were used to acquire local field potential (LFP) and multi-unit spike activity (MUA) through the depth of LGN and VC (*Figure 1A,B*). Whole-field visual stimulation was used to identify L4 in VC and eye-specific lamina in LGN. The LGN channel with maximal contralateral visual response was used for analysis. The VC and LGN recordings were not topographically aligned. This allows investigation of the oscillations in each structure reported here, but not precise phase relationships between LGN and VC during spontaneous activity.

As previously described (*Colonnese and Khazipov, 2010*), spontaneous activity in VC during the retinal wave period consisted of long periods of network silence interrupted by periods of activity lasting 3–10 s. Activity within these active periods consisted of spindle-shaped oscillations in the local field potential (LFP) in superficial layers with a primary frequency of 20.8 ± 0.8 Hz (n = 7, mean ± SEM) and elevated MUA in all layers (*Figure 1D,H*). Activity in LGN had a similar structure: 5–10 s periods of spiking activity alternating with periods of low or absent spiking (*Figure 1C*). Because the amplitude of the LGN LFP was lower than the median spike amplitude, it was not analyzed further. To determine if the spindle-burst oscillations observed in VC originate in LGN, we quantified the frequency modulation of LGN MUA. Spike-rate modulation in LGN showed a primary frequency of 23.1 ± 0.6 Hz (n = 7) (*Figure 1G*), similar to VC. Thus, spindle-burst oscillations are also observed in LGN. Combined with the absence of significant oscillatory ganglion cell firing during waves (*Colonnese and Khazipov, 2010*), and the location of the VC spindle-bursts' current sink in the input layer, this data supports a thalamic origin of spindle-burst oscillations.

### Spindle-burst oscillations require retina and thalamus

To quantify the contribution of retinal input to spontaneous activity in LGN (*Figure 2A*) and VC (*Figure 2B*), we acutely silenced both eyes by intraocular injection of the blood brain barrier-impermeant glutamate receptor antagonists APV and CNQX. Silencing was confirmed by loss of the visual response (not shown). As predicted, retinal silencing dramatically reduced firing rates in LGN (–86.6 ± 3.9%, mean ± SEM) and VC (–67.2 ± 10.2%). The continuity of activity, measured as the proportion of the recording with spike activity (periods containing at least two spikes with an interval of less than 500 ms, called 'events'), was also significantly reduced (LGN –88.5 ± 6.5%, VC –90.6 ± 2.3%). Examination of the distribution of event lengths after retinal silencing showed that events over 5 s in duration were reduced in LGN and eliminated in VC (*Figure 2A4,B4*). Retinal silencing largely eliminated the prominent peak in LGN MUA and VC LFP spectra in the spindle-burst range, leaving only lower frequency activity (*Figure 2A5,B5*). In combination with previous results showing that spindle-burst oscillations are driven by retinal waves (*Colonnese and Khazipov, 2010*), the present retinal silencing data confirms that in LGN, as in VC, spontaneous retinal activity is the primary driver of spontaneous activity and spindle-burst oscillations. In the absence of retinal input, spontaneous activity in the intact thalamocortical loop can provide at most 20% of normal LGN firing.

To determine the ability of VC alone to generate spontaneous activity and spindle-burst oscillations, we pharmacologically silenced LGN by local injection of the GABA receptor agonist muscimol (*Figure 2C*). Silencing LGN significantly reduced VC L4 MUA rate (–79.0 ± 3.9%) and continuity (–89.0 ± 2.6%) (*Figure 2C3*). The remaining VC activity consisted only of very short bursts (<1 s) (*Figure 2C4*). Most importantly, LGN silencing completely eliminated oscillatory activity in the VC LFP (*Figure 2C5*). All of these activity reductions were more severe than observed following retinal silencing; thus while VC alone can generate very short bursts of activity, as observed in slices (*Allène et al., 2008*), an intact thalamocortical loop is necessary for the generation of activity lasting longer than 1 s that contains any high frequencies. In total these data confirm and extend

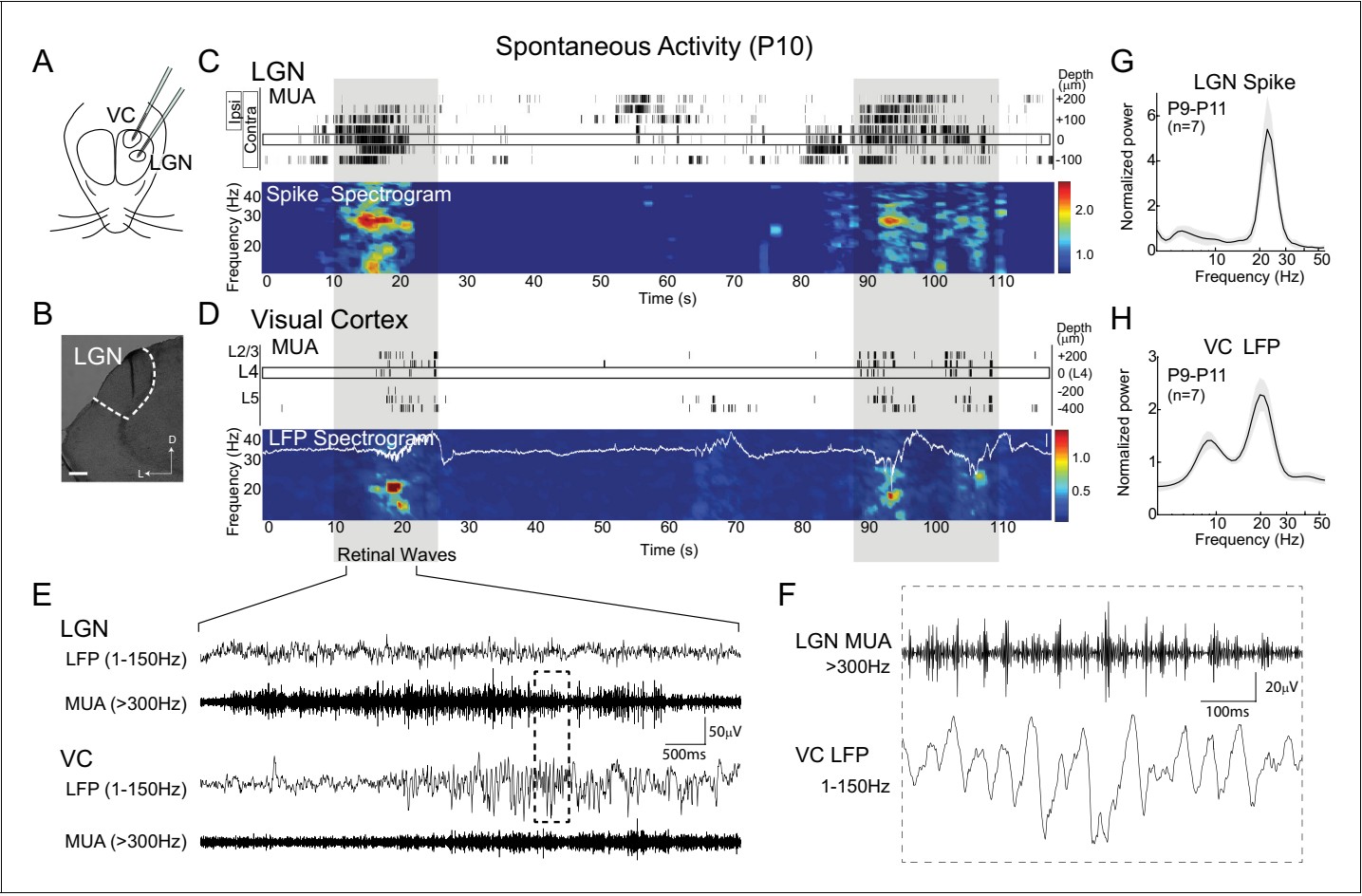

**Figure 1.** Thalamic spindle-burst oscillations are transmitted to visual cortex. (**A**) Experimental setup. Simultaneous recordings from visual thalamus (lateral geniculate nucleus, LGN) and primary monocular visual cortex (VC) were acquired with two single shank multi-electrode arrays in awake, head-fixed rats. (**B**) Reconstructed electrode track in LGN. Scale bar: 100 μm (**C**) Representative spontaneous activity in LGN of a P10 rat. Raster plot shows multi-unit spiking (MUA) from contacts separated by 50 μm. Eye specificity of each channel is shown by box on left. Spike spectrogram shows frequency of MUA modulation for LGN channel with maximal contralateral visual response (rectangle at 0 μm). Long-duration spiking events previously shown to result from retinal waves (*Colonnese and Khazipov, 2010*), marked by grey shading, are associated with elevation in beta-band frequencies in LGN and VC, referred to a spindle-burst oscillations. (**D**) Representative spontaneous activity in VC simultaneously recorded with (**C**). Raster plots show MUA at multiple depths. Local field potential (LFP) recording in L4 is plotted on LFP-derived spectrogram. Elevations in beta-band frequencies (spindle-bursts) are associated with similar activities in LGN. Scale bar is 200 μV. (**E**) LFP (1–150 Hz) and MUA (>300 Hz) in LGN and VC during the retinal wave shown in (**C**) and (**D**) at expanded time scale. (**F**) LGN MUA (>300 Hz) and VC LFP (1–150 Hz) during a retinal wave shown in (**E**) at expanded time scale. (**G**) Population mean of normalized LGN spike spectra for P9-P11 (n = 7). Grey shading is standard error of the mean. (**H**) Population mean of normalized VC LFP spectra for P9-P11 (n = 7).

previous results in somatosensory (*Minlebaev et al., 2011*; *Yang et al., 2013a*) and visual (*Colonnese and Khazipov, 2010*) cortex that spindle-burst oscillations first emerge in thalamus in response to retinal or sensory input.

## Transmission of retinal waves requires corticothalamic feedback

The thalamic and cortical activity remaining after retinal and LGN silencing suggests that corticothalamic excitation is at least partially functional in neonatal rats, as has been previously demonstrated in ferrets (*Weliky and Katz, 1999*). We therefore tested the hypothesis that VC provides critical amplification and transformation of the retinal input in LGN by silencing VC with local application of APV and CNQX (*Figure 3*). Silencing VC at P9-11 during the peak of stage III waves reduces LGN spike-rate by 79.2 ± 5.6% and continuity by 68.4 ± 9.3% (*Figure 3D2*). Remarkably, these activity

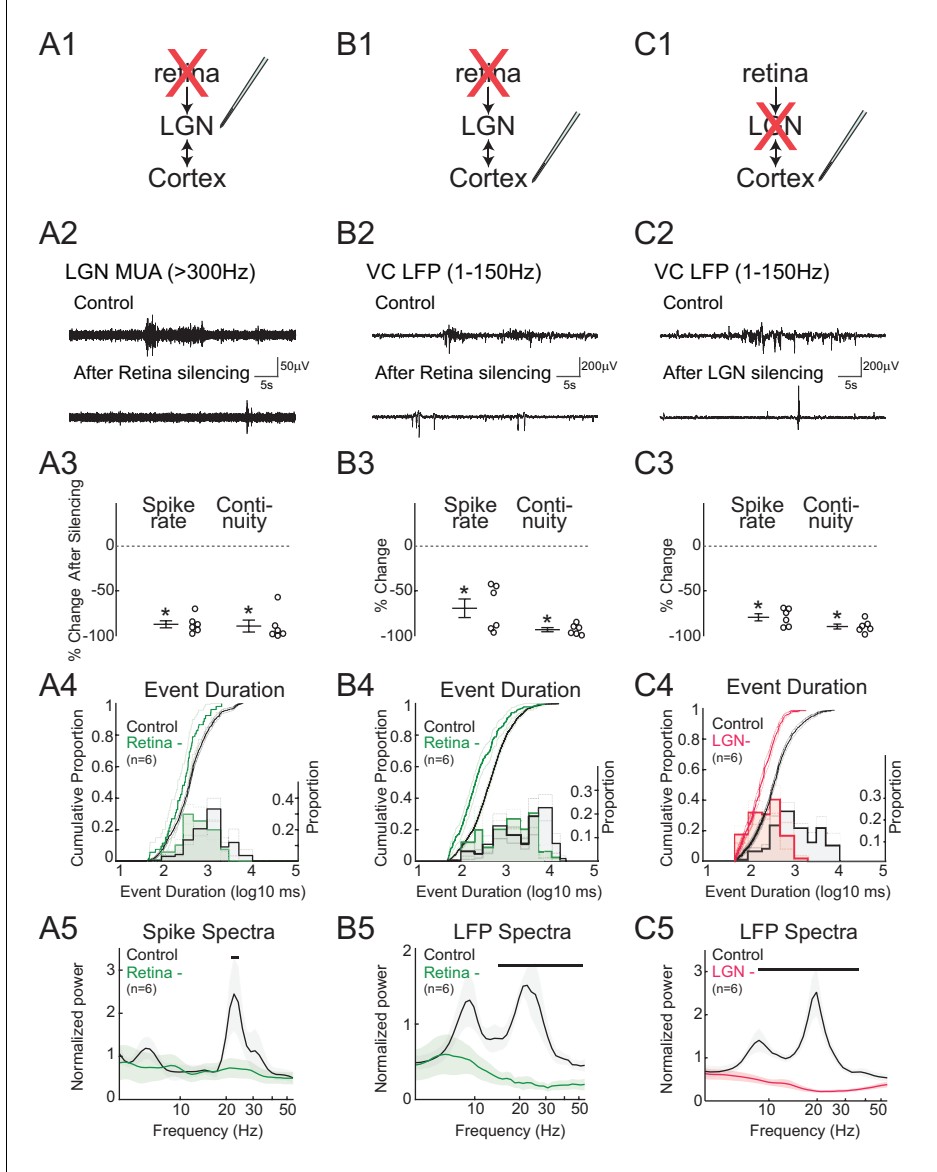

**Figure 2.** Spindle-burst oscillations require retinal and thalamic activity. (**A**) Blockade of spontaneous retinal waves reduces thalamic firing and spindle-burst oscillations. (**A1**) Ocular injection of glutamate receptor antagonists APV and CNQX to silence retinal activity with simultaneous recording in LGN. (**A2**) Representative LGN MUA (>300 Hz) before (control) and after retinal silencing. (**A3**) Percent change in multi-unit spike rate and continuity of activity during retinal blockade (n = 6, Wilcoxon signed-rank test for difference from pre-silence, p=0.0313, p=0.0313). Spike continuity is calculated as the proportion of periods containing at least two spikes with interval of less than 500 ms. (**A4**) Analysis of event duration. Cumulative distribution is shown in solid lines (left y-axis). 95% confidence interval (CI) is shown as shaded area (n = 856 and n = 52, two sample KS test, p=0.0193). Proportion distribution is additionally plotted as a bar graph to aid visualization of event duration (right y-axis). (**A5**) Population mean of normalized spike spectra. Silencing retina reduced spindle-burst frequencies (n = 6, permutation test, p<0.05 between 23.9–25.5 Hz, bar = frequencies significantly different by permutation test). (**B**) Silencing retinal activity reduces cortical firing and spindle-burst oscillations. (**B1–5**) As for **A1–5** but for L4 of VC (**B2**: VC LFP (1–150 Hz); **B3**: n = 6, Wilcoxon signed-rank test, p=0.0313, p=0.0313; **B4**: n = 1109 and 189, two sample KS test, p=$10^{-12}$; **B5**: n = 6, permutation test, p<0.05 between 14.4–54.6 Hz). (**C**) Silencing thalamic activity with muscimol while recording in VC. (**C1–5**) As for **B1–5**. Larger reduction in spontaneous firing, event duration and spectral power than for retinal silencing suggests residual activity in thalamocortical loop after retinal silencing can generate small amounts of spindle-burst activity. (**C2**: VC LFP (1–150 Hz); **C3**: n = 6, Wilcoxon signed-rank test, p=0.0313,

*Figure 2 continued on next page*

*Figure 2 continued*

p=0.0313; **C4**: n = 811 and 377, two sample KS test, p=10$^{-22}$; **C5**: n = 6, permutation test, p<0.05 between 8.6–42.5 Hz). All error bars are SEM unless noted.

reductions are of similar magnitude to those caused by retinal silencing. The fact that silencing of either retina or VC results in a greater than 50% reduction in firing indicates that they act synergistically to drive LGN. One explanation is that VC participates in feedback amplification as part of an excitatory loop with LGN that is necessary for the generation of effective thalamic spiking during retinal waves. The effect of VC silencing was limited to the LGN; spike-rates for all contacts from the same shank located 20–100 μm below the last visually responsive contact were not affected (+3.1 ± 17.1%, p=1.000, n = 6, Wilcoxon signed rank test), indicating that the pharmacological silencing of VC did not have systemic effects. Event duration was reduced, but not as severely altered as by retinal silencing (*Figure 3D5*), indicating that retina initiates and maintains activity in LGN, but VC prolongs it. Silencing VC strongly reduced normalized spike spectral power in the spindle-burst range (*Figure 3D3*). VC input appears to be required for normal patterning of the spindle-oscillations, as peak frequency of remaining activity shifted from 24.6 ± 0.9 to 16.2 ± 2.0 Hz (*Figure 3D4*) after VC silencing. These results show that during the period of peak stage III retinal waves, feedback connections from VC to LGN are a critical component of retinal wave transmission, which serve to both amplify and prolong spontaneous activity as well as accelerate spindle-burst oscillations in LGN.

Retinothalamic innervation precedes cortical innervation of the thalamus (*Seabrook et al., 2013*). To determine whether corticothalamic feedback is also critical for the stage II retinal waves (cholinergic) present during the first post-natal week, we examined the effect of VC silencing at P5-7 (*Figure 3C*). Spindle-burst oscillations are slower at this age, resulting in a peak frequency of the L4 LFP spectra of 12.2 ± 1.7 Hz (data not shown). Spontaneous LGN spike-rate frequency is similarly down-shifted (16.0 ± 1.2 Hz). Silencing VC at this age significantly reduced spike-rates (−52.0 ± 7.4%) and continuity (−49.8 ± 10.7%) (*Figure 3C2*), showing that corticothalamic amplification is present at this age as well. However, there was no significant effect of VC silencing on either spike spectra or event duration (*Figure 3C3–5*). Thus during the first post-natal week, the role of VC is limited to amplification. At this age, LGN is able to generate spindle-oscillations in response to retinal waves, albeit with a lower frequency than during the second postnatal week. The difference between P5-7 and P9-11 demonstrates a shift in the role of corticothalamic feedback from simple amplifier to active participant in the shaping of LGN oscillations.

Spontaneous activity in VC dramatically changes between P11 and P13. Cortex no longer exhibits the long silent periods of immaturity; activity becomes continuous and loses the prominent spindle-oscillations, which are replaced by adult-like, broad-band activity (*Colonnese and Khazipov, 2010*). To determine if the LGN requirement for corticothalamic feedback is restricted to the period of retinal waves, and what role VC plays in the maturation of spontaneous activity, we silenced VC while recording from LGN at P13-14. VC silencing at this age does not decrease LGN activity, but rather increases spike rates (+26.2 ± 5.4%) and there is slight but not significant increase in continuity (+13.5 ± 5.0%) (*Figure 3E2*). VC silencing increases the proportion of very long duration events (>10 s) (*Figure 3E5*), but does not change the LGN MUA spectrum (*Figure 3E3*). Thus the excitatory role of corticothalamic feedback is limited to the period of retinal waves, and cortex becomes inhibitory to thalamus around eye-opening.

Together the VC silencing experiments indicate a continually changing role for corticothalamic feedback in the transmission of retinal activity during development. During the period of retinal waves, VC essentially gates LGN output, first as a simple amplifier of LGN, and later in the second post-natal week, by amplifying and patterning oscillations in LGN. This amplification role ends before eye-opening, after which corticothalamic feedback plays a less prominent role in the generation and patterning of retinal transmission through thalamus.

## Absence of inhibition facilitates corticothalamic feedback excitation

To determine the mechanism of early corticothalamic feedback excitation and rhythmogenesis, we took advantage of the fast kinetics and high sensitivity of the blue light drivable channelrhodopsin

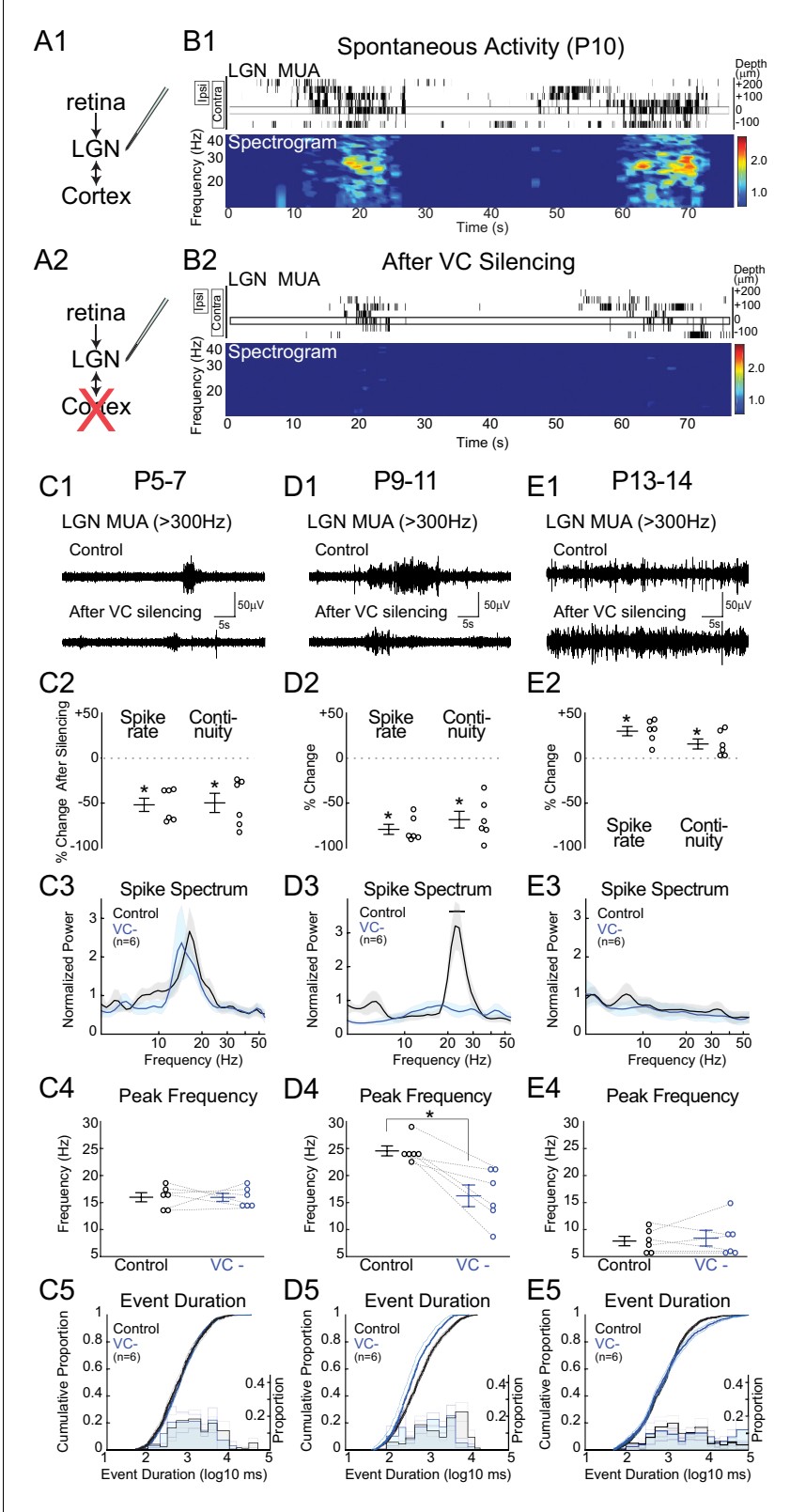

**Figure 3.** Corticothalamic feedback amplifies retinal waves in thalamus. (**A**) Experimental set-up: silencing VC with local application of APV and CNQX while recording in LGN. (**B**) Representative spontaneous activity in LGN before (**B1**) and after VC silencing in a P10 rat (**B2**). LGN MUA rasters and spike-rate spectrogram as for *Figure 1*. (**C1**) Representative LGN MUA (>300 Hz) before (control) and after VC silencing at P5-7. (**C2**) Percent change in multi-

*Figure 3 continued on next page*

*Figure 3 continued*

unit spike rate and continuity of activity after VC silencing during cholinergic retinal waves (P5-7). VC silencing reduces LGN spiking as early as P5 (n = 6 each, Wilcoxon signed-rank test for difference from pre-silencing, p=0.0313, p=0.0313). (C3) Population mean of normalized LGN spike spectra. Note no change in frequency distribution despite decreased spike rates at P5-7 (permutation test, p>0.05 at all frequency ranges). (C4) Peak of spike frequency distribution (Wilcoxon signed-rank test, p=0.8750). (C5) Event duration distributions before and after VC silencing (n = 1727 and 728, two sample KS test, p=0.1975). (D) Effects of VC silencing during glutamatergic waves (P9-11) (n = 6). (D1–5) as for C1–5 (D2: Wilcoxon signed-rank test, p=0.0313, p=0.0313; D3: permutation test, p<0.05 between 19.8–27.2 Hz; D4: Wilcoxon signed-rank test, p=0.0313; D5: n = 1549 and 773, two sample KS test, p=$10^{-9}$). Note large decrease in spindle-oscillation power and peak frequency in LGN following VC silencing. (E) Effects of VC silencing after retinal wave period (P13-14) (n = 6). (E1–5) as for C1–5 (E2: Wilcoxon signed-rank test, p=0.0313 p=0.0313; E3: permutation test, p>0.05 at all frequency ranges; E4: Wilcoxon signed-rank test, p=0.8750; E5: n = 1785 and 1129, two sample KS test, p=0.0029). Note absence of effect on spike spectra, and increase, in contrast to the decreases until P11, in spike rate, continuity and event duration in LGN following VC silencing at P13-14.

Chronos (*Klapoetke et al., 2014*) to gain control of VC neurons within a few days of birth. Injection of AAV-Chronos-GFP into VC at P0-1 yielded strong expression in VC neurons in all layers and axon terminals by P6. In neonatal mice (*Seabrook et al., 2013*; *Grant et al., 2016*) cortical axons avoid LGN relative to surrounding thalamic nuclei. We observed the same pattern in VC axons expressing Chronos-GFP (*Figure 4B*), although cortical axon arbors can be seen within LGN even at P6. A 10 ms pulse of 470 nm LED on VC was sufficient to activate deep VC layers at all ages with similar delay (P6-7 3.7 ± 1.8 ms; P9-11 3.4 ± 1.5 ms; P13-14 3.5 ± 1.3 ms). At all ages, 10 ms light-pulses onto VC evoked short duration (~30 ms) initial spiking in LGN (*Figure 4C,D*), although the latency to activate LGN following VC activation decreased between P6-7 (34.2 ± 2.7 ms) and P9-11 (21.5 ± 0.5 ms), stabilizing by P13-14 (20.0 ± 0.9 ms) (*Figure 4E*). The probability of generating a spiking response in LGN with each VC activation increased from 39.7 ± 7.0% at P6-7 to 92.4 ± 3.6% at P13-14 (*Figure 4F*). The spike rate increase was similar in all three ages (*Figure 4G*).

The major developmental change in the LGN response occurred between P9-11 and P13-14. It consisted of the development of post-stimulus inhibition following the initial evoked response (*Figure 4H*). The relative increase in LGN spike-rates at 100–250 ms after VC stimulation switched from neutral or positive (P6-7 −0.1 ± 36.7%; P9-11 +63.4 ± 61.0%) to negative at P13-14 (−89.2 ± 8.8%) indicating the recruitment of functional feedforward inhibition into the corticothalamic circuit between P11 and P13.

To determine how the increasing speed, reliability and inhibition of corticothalamic feedback could contribute to the ability of VC to generate spindle-burst oscillations at P9-11, but not P5-7 or P13-14, we examined the LGN response to patterned VC stimulation at these ages. Five 10 ms light pulses at 20 Hz evoked different responses in LGN at each age (*Figure 4I*). At P6-7, 20 Hz VC stimulation increased LGN spike rates but did not induce oscillatory activity. In contrast, the same stimulation at P9-11 induced 20 Hz oscillatory firing in LGN. At P13-14, despite reliable LGN responses to the first one or two pulses, induced inhibition prevented further oscillation.

Together these cortical stimulation experiments provide a mechanistic explanation for the changing role of corticothalamic feedback in shaping LGN activity. During cholinergic retinal waves, connectivity is sufficient for VC to amplify retinal waves in LGN but its reliability and velocity are insufficient to synchronize LGN firing. Then during the second postnatal week when glutamatergic waves predominate (P9-11), increasing speed and reliability of the feedback loop enables cortex to synchronize LGN at higher frequencies than LGN alone would naturally oscillate. Finally, at the transition to the third week (after P13), VC afferents recruit an inhibitory circuit, which reduces their ability to synchronize activity at high frequencies and makes the net effect of corticothalamic feedback inhibitory, as in the adult.

## Corticothalamic feedback excitation specifically amplifies spindle-burst oscillations

During the period of stage III retinal waves, visual responses can be evoked in retina (through closed eyelids) and these are transmitted to VC (*Colonnese et al., 2010*). Until just before eye-opening,

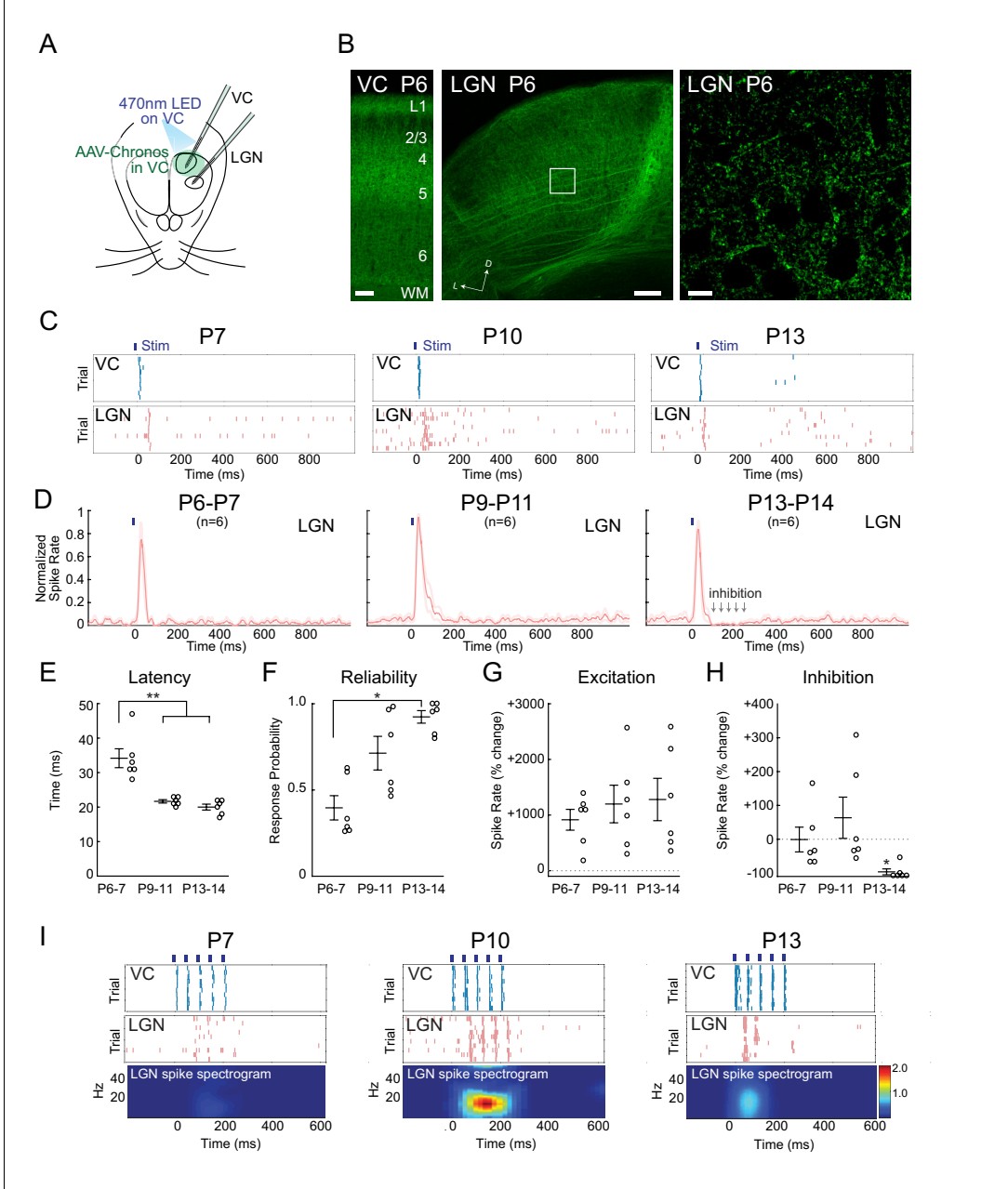

**Figure 4.** Increasing reliability and recruitment of inhibition transforms corticothalamic function. (A) Experimental setup: Optogenetic stimulation of VC neurons expressing Chronos-GFP while simultaneously recording in LGN and VC. AAV-Chronos-GFP was injected into VC at P0-1. (B) Representative images of Chronos-GFP expression in the VC and Chronos-GFP expressing corticothalamic axons in LGN at P6. Scale bars: 100, 100 and 10 μm. (C) Representative MUA rasters for 10 trials with LGN responses at each age. (D) Population mean post-stimulus time histogram of LGN spiking (n = 6 for each age group). At P6-11 VC optogenetic stimulation evokes purely excitatory responses in LGN, while in P13-14 animals initial excitation is followed by inhibition. (E) Population mean delay between LGN and VC spike onset (Kruskal-Wallis test, P6-7 vs P9-11 p=0.0020, P6-7 vs P13-14 p=0.0058, P9-11 vs P13-14 p=0.6247). (F) Population mean reliability: percentage of trials in which VC spike activity produced LGN spike activity (Kruskal-Wallis test, p=0.1744, 0.0057, 0.3960). (G) Population mean of spike-rate change in LGN following VC stimulation (1–100 ms after stim) (Wilcoxon signed-rank test vs baseline, P6-7: p=0.0313, P9-11: p=0.0313, P13-14: p=0.0313). (H) Population mean spike-rate change in LGN during inhibitory period (100–250 ms after stim). Significant inhibition is observed only at P13-14 (Wilcoxon signed-rank test vs baseline, P6-7: p=0.6563, P9-11: p=0.8438, P13-14: p=0.0313). (I) Cortical stimulation at spindle-burst frequencies (20 Hz) induces oscillations in LGN only at P9-11. VC stimulation at 20 Hz caused firing in LGN at all ages, but entrained LGN only at P9-11.

visual responses have an immature structure that consists of two distinct phases: an initial response composed of 'early gamma oscillations' (EGOs) (*Khazipov et al., 2013b*) followed by a secondary response consisting of spindle-bursts (*Colonnese et al., 2010*). We used this pattern to examine the differential role of corticothalamic feedback on these two prominent early activity patterns, as well as in the transition to the adult pattern. P9-11 LGN visual responses also consisted of early-gamma (30–50 Hz) followed by spindle-burst oscillations (8–30 Hz) (*Figure 5B*). Silencing VC reduced spike rates in LGN during both the primary and secondary responses (−46.2 ± 4.8% and −48.7 ± 5.8% respectively) (*Figure 5D,F*), but reduced the power of spike-rate oscillations only for spindle-burst oscillations (*Figure 5G*). These results confirm the strong amplification role of early corticothalamic feedback, and specifically implicate corticothalamic excitation in the generation of spindle-bursts, but not EGOs.

At P13-14 visual responses in LGN were shorter and consisted of a rapid, non-oscillatory primary response, and greatly reduced secondary response (*Figure 5C*). This is identical to changes observed in VC (*Colonnese et al., 2010*). As observed for spontaneous activity, silencing VC at this age no longer reduced LGN activity. Instead, it caused a non-significant increase in the evoked spike-rate during the primary response (+15.1 ± 13.3%) and a significant increase in the secondary response (+48.5 ± 9.0%) (*Figure 5F*). These results show that the switch in corticothalamic feedback from functionally excitatory to inhibitory is not limited to spontaneous activity. They further show that corticothalamic feedback is not necessary for the loss of evoked gamma and spindle-bursts from visual responses at P13, which instead appear to depend on altered circuitry in LGN and retina.

## Discussion

We used pharmacological silencing and optogenetic stimulation in unanesthetized rats in vivo to show a change in the functional role of corticothalamic projections during development. Before vision, when spontaneous retinal waves instruct circuit formation (*Huberman et al., 2008*), the corticothalamic projection provides excitatory feedback amplification to thalamus. Although retina initiates activity in central visual circuits, retinal waves alone provide as little as 20% of the total drive. Corticothalamic feedback amplification accounts for the remaining 80%, effectively gating transmission of retinal waves in thalamus. During the second post-natal week corticothalamic excitation not only amplifies, but also synchronizes, thalamic firing, effectively conditioning its own input. Our results show that the transmission of early spontaneous activity is not simply feed-forward, and that feedback circuits can have specialized developmental properties that ensure the reliable transmission of early activity from the periphery to the cortex. The mechanism is the delayed development of inhibitory recruitment by the corticothalamic projection. This is a common observation in developing circuits (*Le Magueresse and Monyer, 2013*), yet the functional significance has been difficult to prove.

### Feedback amplification provides the primary thalamic drive during retinal waves

The present work reinforces previous findings (*Weliky and Katz, 1999*; *Colonnese and Khazipov, 2010*; *Ackman et al., 2012*) that sustained (>2 s) thalamocortical activity before eye-opening is driven by central transmission of retinal waves. Neither VC alone, nor an intact LGN-VC loop, is capable of producing more than brief bursts of activity. In contrast, without VC, firing in LGN is dramatically reduced but sustained activity is still present. Only the longest events—caused by sustained firing between waves occurring in a cluster (*Blankenship and Feller, 2010*)—are suppressed by VC silencing, demonstrating that corticothalamic feedback maintains activity during waves but does not initiate activity. The fact that VC and retina are both necessary for over 50% of LGN spiking shows their drive is multiplicative, not additive. Together our findings suggest that VC makes a feed-forward excitatory loop with thalamus that amplifies the drive from retina. In ferrets corticothalamic feedback synchronizes activity between eye-specific lamina (*Weliky and Katz, 1999*), but its role in amplification and oscillation generation is unknown, as is whether this function is unique to rodents.

In addition to amplifying retinal activity, central circuits transform ganglion cell firing during waves into spindle-burst oscillations (*Colonnese and Khazipov, 2010*). Almost all activity in early VC, as well as every other cortical region examined, occurs as rapid oscillations necessary for synaptic stabilization in the developing circuit (*Khazipov et al., 2013a*, *2013b*). Determining the similarities and

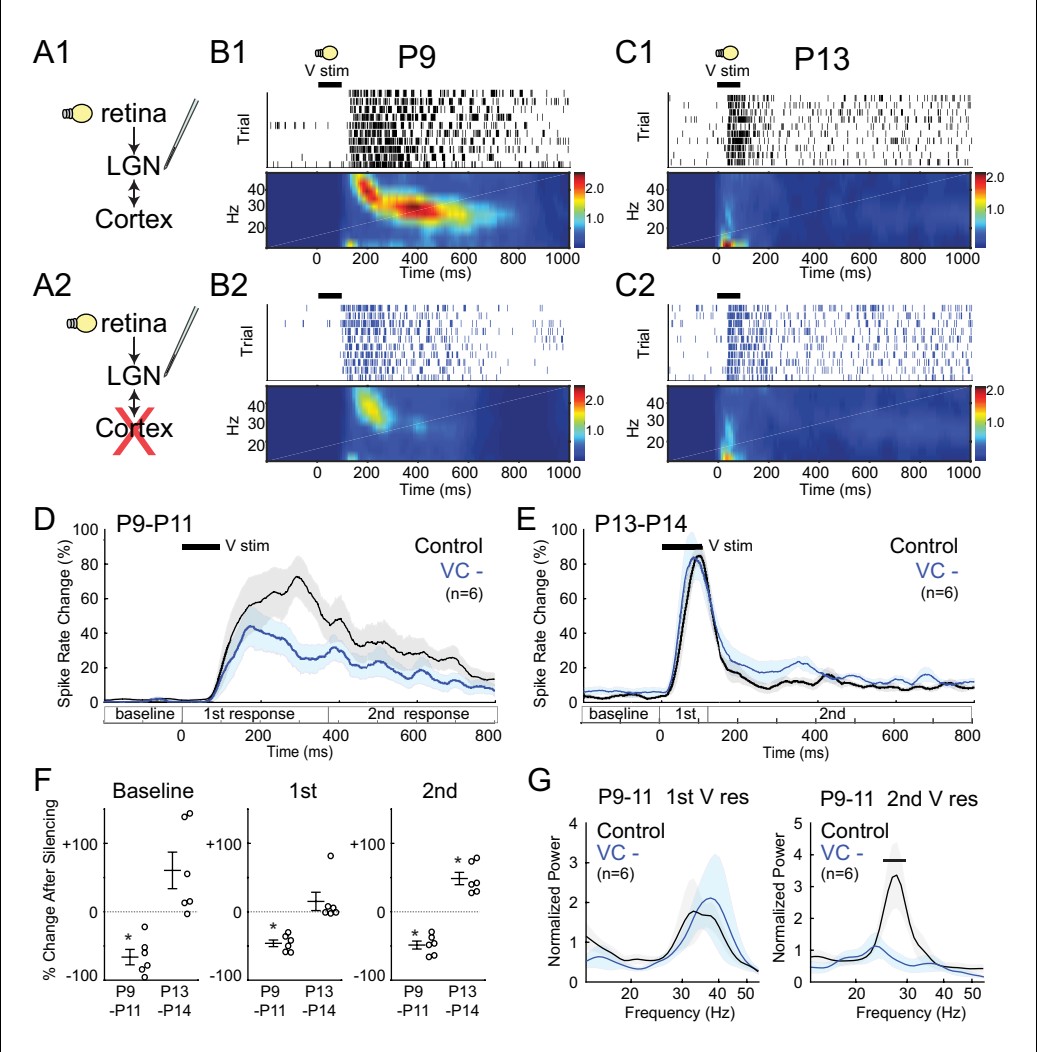

**Figure 5.** Corticothalamic feedback required for spindle-burst oscillations but not early-gamma oscillations. (**A**) Visually evoked MUA responses were measured in LGN before (**A1**) and after (**A2**) VC silencing. Visual stimulus was 100 ms flash. (**B**) Representative evoked LGN MUA before (**B1**) and after VC silencing (**B2**) in a P9 rat. Before VC silencing, visual responses consists of an initial (first) early-gamma oscillation (EGO), followed by a spindle-burst (second), as observed in VC (*Colonnese et al., 2010*). After VC silencing, EGOs remain, but spindle-bursts oscillations are reduced. (**C**) Representative visual responses at P13 before (**C1**) and after VC silencing (**C2**). Note disappearance of EGOs and spindle-burst in LGN after P13 as observed in VC (*Colonnese et al., 2010*). (**D**) Population mean post-stimulus spike rate histogram for P9-11 (n = 6). Spike rates are reduced by VC silencing both during primary (EGO) and secondary (spindle-burst) responses. (**E**) Population mean post-stimulus spike rate histogram for P13-14 (n = 6). LGN spike-rates are increased by VC silencing at P13-14, in contrast to decrease after VC silencing at P9-11. (**F**) Population mean change in LGN spike-rate caused by VC silencing in the baseline (left), primary visual (first, middle) and secondary visual (second, right) responses for both age groups. (P9-11, n = 6, Wilcoxon signed-rank test for difference from pre-silencing, baseline: p=0.0313, first: 0.0313, second: 0.0313; P13-14, n = 6, p=0.0625, 0.4375, 0.0313). (**G**) Normalized LGN spike spectra for first (left) and second (right) visual response at P9-11. Despite reductions in spike rate for both responses, only spindle-burst frequencies (second) are reduced by VC silencing (n = 6, bar = frequencies significantly different by permutation test, first: p>0.05 at all frequency ranges, second: p<0.05 at 26.7–31.7 Hz).

differences between the cortical areas in how these rhythms are generated will inform our understanding of how regional specialization develops. In all areas, rapid oscillations occur in response to spontaneous and evoked firing of the dominant input (eg. whisker in barrel cortex), with a frequency inversely proportional to its temporal and spatial specificity (*Khazipov et al., 2013b*). Here we identify LGN as the locus of generation of VC spindle-bursts, similar to somatosensory (VPN) thalamus (*Yang et al., 2013a*). Unlike VPN, corticothalamic feedback plays a critical role for setting the frequency and amplitude of LGN firing. During the first post-natal week, corticothalamic connections are capable of directly driving LGN, but only at long latency and with poor reliability, consistent with their low density in LGN at this age (*Seabrook et al., 2013*; *Grant et al., 2016*), and they are not necessary for rhythmogenesis. How spindle-bursts are generated in LGN is unknown, but likely involves a combination of functional connectivity with the thalamic reticular nucleus (*Evrard and Ropert, 2009*) and intrinsic excitability of relay neurons (*Lo et al., 2002*). Between the first and second post-natal weeks, roughly synchronized to the switch between cholinergic and glutamatergic waves (*Huberman et al., 2008*), the timing and reliability of corticothalamic connectivity improves, changing its role from simple amplification to synchronization and rhythmogenesis. LGN can still generate its own weak (15 Hz) oscillations in the first week, but it requires VC to generate normal (25 Hz) activity in the second week. Because corticothalamic projections are only capable of synchronizing LGN at these frequencies at this age (*Figure 4*), and L5/6 neurons are synchronized by the spindle-oscillations (*Colonnese and Khazipov, 2010*), we suggest that the faster spindle-burst oscillations are generated in the feedback circuit between the two structures.

The role of corticothalamic feedback appears specific to spindle-bursts, as it is not important for the generation of the other major early oscillation, Early-Gamma Oscillations (EGOs). In contrast to spindle-bursts, EGOs activate only the input and superficial layers of a single column, and occur only in response to temporally and spatially limited input (*Khazipov et al., 2013b*), for example in response to a single deflection of the principle whisker in barrel cortex. VC EGOs do not occur in response to retinal waves, but can be induced by visual stimulation (*Colonnese et al., 2010*), allowing us to examine their cortical dependence. While LGN spiking during the period of EGOs was reduced by VC silencing, the normalized oscillation power was unaffected, suggesting that VC provides amplification but not rhythmogenesis for EGOs. Poor modulation of EGOs by corticothalamic projections is consistent with the conduction delays (>20 ms) and lack of L5/6 firing during EGOs (*Colonnese et al., 2010*) which should prevent corticothalamic projections from contributing to such rapid synchronization of LGN. In barrel cortex, cortical silencing reduces frequency power for EGOs but not spindle-bursts (*Yang et al., 2013a*). Thus corticothalamic feedback may be differentially tuned by region, as EGOs occur spontaneously in barrel-cortex but not in VC (*Yang et al., 2009*; *Cichon et al., 2014*).

## A transient circuit specialized for transmission of retinal waves

Finding the circuit specializations that allow for transmission of activity during early development will help determine genetic and environmental causes of neurological disorders (*Kanold and Luhmann, 2010*). Defective transient circuits can disrupt circuit formation and then disappear, leaving no trace in the adult. Failure to reorganize early circuits into their mature configurations is another likely cause of adult dysfunction (*Ben-Ari, 2008*). Important transitional synaptic properties include elevated input resistance, long duration NMDA currents with poor $Mg^{++}$ block, glutamate spillover, and plateau potentials (*Lo et al., 2002*; *Hauser et al., 2014*). Transitional circuits include early-born subplate neurons and L5 somatostatin neurons, both of which receive early thalamocortical synapses (*Kanold and Luhmann, 2010*; *Marques-Smith et al., 2016*; *Tuncdemir et al., 2016*).

Here we show clear evidence in vivo that a transient circuit outside the retina (*Blankenship and Feller, 2010*) participates in the transmission of early activity. Amplification involves functional strength that seems larger than that predicted given VC terminal density (*Seabrook et al., 2013*; *Grant et al., 2016*), but also the late integration of inhibition into the circuit. The source of corticothalamic feedback could include contributions from the normal L6 projection as well as L5, whose axons traverse LGN without extensive arborization due to competition from retinal axons (*Grant et al., 2016*). The capacity of these axons to sprout following retinal blockade suggests that axon terminals are present.

Our results identify corticothalamic feedback as a potential contributor to circuit refinements that are dependent on retinal waves in LGN and cortex (*Huberman et al., 2008*). The amplification

identified here likely contributes to the defects in retinal ganglion cell targeting caused by genetic ablation of cortex (*Shanks et al., 2016*). We predict that less dramatic manipulations such as silencing corticothalamic feedback should also disrupt refinement of retinotopy and other receptive field properties at eye-opening in LGN and VC. However, the inability to specifically block corticothalamic feedback, without disrupting the intra-cortical connections of L5/6 circuits, make definitive experiments unfeasible. The prominent role of corticothalamic feedback in patterning LGN responses to glutamatergic waves suggests it is particularly critical for synaptic refinement driven by these waves. The role of glutamatergic waves remains controversial (*Kerschensteiner, 2016*), but their perturbation reduces segregation of C-lamina and maintenance of A-lamina segregation in LGN (*Davis et al., 2015a*, *2015b*) as well as orientation selectivity in VC (*Huberman et al., 2008*).

One of the most surprising roles for corticothalamic feedback is the acceleration of spindle-oscillations during late pre-visual development. A similar near doubling of occipital delta-brush frequencies occurs in human infants between 27 and 35 gestational weeks (*André et al., 2010*). Thus, acceleration of early oscillations is a conserved feature of cortical development, for which we have identified a mechanism. Why does frequency increase? Driving cortical inputs at 10 Hz causes synaptic depression in immature somatosensory cortex at age equivalents (*Minlebaev et al., 2011*), while higher frequencies cause potentiation. Thus the acceleration in oscillation frequency may be a function of the change from 'burst-based' plasticity rules used to grossly align topography (*Butts et al., 2007*), to more precise spike-timing dependent plasticity that can take advantage of the increasing specificity of firing patterns within retinal waves (*Kerschensteiner and Wong, 2008*) to reduce poly-innervation of relay cells (*Jaubert-Miazza et al., 2005*) and refine orientation selectivity (*Huberman et al., 2008*).

Overall our results show that thalamocortical circuits function as an integrated circuit even during early development, and suggest that spontaneous retinal activity influences the development of corticothalamic feedback to LGN and reticular thalamus (RT).

## Transient feedback excitation is terminated by development of corticothalamic inhibition

This is a systematic investigation of the functional development of a 'modulator' (*Sherman and Guillery, 2009*) feedback pathway in vivo. Previous investigations have focused on driver inputs, such as retinogeniculate and thalamocortical synapses (*Ackman and Crair, 2014*). For drivers, feedforward inhibition develops later than excitation (*Ziburkus et al., 2003*; *Colonnese, 2014*), but their functional effect remains excitatory. Here we show that for a modulator its role during development is different from its role in the adult, switching from an excitatory and amplifying role early, to a modulatory, net-inhibitory role later. This principle may apply to other modulatory feedback projections, such as intra-cortical feedback, which largely synapse on interneurons and develop later than feedforward connections (*Berezovskii et al., 2011*; *Yang et al., 2013b*). It should be noted we used coarse assays of corticothalamic function which, while sufficient to reveal developmental differences in this pathway, reveal little about its adult function, which appears to have excitatory or inhibitory effects that are strongly dependent on stimulus and behavioral state (*Denman and Contreras, 2015*; *Crandall et al., 2015*).

The lack of thalamocortical feedback inhibition has implications for seizure susceptibility during infancy, because thalamocortical synchronization through RT is an important mechanism of seizure spread (*Paz et al., 2013*). Preterm infants and neonates are highly susceptible to seizures (*Rakhade and Jensen, 2009*), and excitatory feedback amplification through thalamus may amplify paroxysmal events. Conversely, lack of inhibitory recruitment by cortex likely contributes to the differences in frequency and propagation of neonatal seizures. Our results suggest that manipulating RT function would be a poor candidate for seizure prevention in preterm populations.

The mechanism by which functional corticothalamic inhibition develops between P11 and P13 likely involves the recruitment of RT, and not LGN, interneurons, as they are poorly driven by VC (*Jurgens et al., 2012*). It is also unlikely to be due to a change in the polarity of GABA$_A$ currents, as GABA is inhibitory in LGN during the first post-natal week. Regardless of the localization of the relevant synaptic potentiation to the cortico-reticular or reticular-LGN synapse, it is clear that the development of effective feedforward inhibition brings to a close the net excitatory feedback loop that amplifies retinal-wave transmission, as well as the ability of cortex to synchronize rapid oscillations within LGN. Feedforward inhibition in both VC and LGN is potentiated between P11 and

P13 (*Lo et al., 2002*; *Colonnese, 2014*), suggesting a common factor is responsible. One prominent candidate for future work includes an increase in continuous activity driven by midbrain neuromodulatory inputs (*Colonnese et al., 2010*) and/or spontaneous non-wave retinal activity.

We have identified a clear developmental role for the late development of inhibition, relative to excitation. A high excitatory to inhibitory ratio during development has been suggested to compensate for low synaptic density (*Le Magueresse and Monyer, 2013*), however a clear correlation with specific developmental events is lacking. By showing a strong contribution to corticothalamic gating of retinal waves, we establish a specific role for late inhibitory development, and provide potential experimental predictions to test its role in circuit formation.

## Materials and methods

### In vivo electrophysiology

All experiments were conducted with approval from The George Washington University School of Medicine and Health Sciences Institutional Animal Care and Use Committee, in accordance with the *Guide for the Care and Use of Laboratory Animals* (eighth Edition, National Academies Press). Long–Evans female rats with litters at postnatal day 4 (P4, birth P0), or pregnant female rats at embryonic day 11–19, were acquired from Hilltop Lab Animals (Scottdale, PA) and housed at one litter per cage on a 12 hr light 12 hr dark cycle. Both males and females were used. No formal sorting or randomization of subjects was applied, and experiments and analysis was not blind to age or treatment. Eyelid opening occurred between P13 and P14. In vivo recording methods are extensively described (*Colonnese and Khazipov, 2010*; *Berzhanskaya et al., 2016*). Topical lidocaine (2.5%) and systemic Carprofen (5 mg/kg) were used as analgesic pre-operatively. To place the headplate, under isoflurane anesthesia (3% induction, 1.5–2% maintenance, verified by toe pinch), the scalp was resected, the skull was cleaned, and a stainless plate with a hole was placed so that the region over occipital cortex was accessible, and the plate was fixed with dental cement. The pups were monitored for signs of stress after recovery from anesthesia. For recording, the animal was head-fixed, and the body was supported within a padded enclosure and warmed with an electric blanket. Temperature was monitored with a thermocouple placed under the abdomen and maintained at 34–36°C. Movement was detected using a piezo-based detector placed under the enclosure. For VC recording, the skull above the monocular primary visual cortex was thinned, and the monocular primary visual cortex was targeted with the following coordinates: 0.5–1.0 mm anterior from the lambda suture, and 2.5–3.0 (P5-7), 2.8–3.3 (P9-11), or 3.0–3.5 (P13-14) mm lateral from *lambda*. Coordinates for LGN were 2.0–2.5 mm anterior and 3.0–3.5 mm lateral (p5-7), 2.3–2.8 mm anterior and 3.3–3.8 mm lateral (p9-11), or 2.5–3.0 mm anterior and 3.5–4.0 mm lateral from the lambda (p13-14). Extracellular activity was recorded using single shank linear multi-electrode arrays (NeuroNexus, MI). A combination of linear 'edge' arrays with 100 or 20 µm separation or 'Poly2' design of two parallel rows with 50 um separation were used. Electrodes were coated with DiI (Life Technologies, CA) before insertion for histological verification of electrode location. In animals older than P8, recording was initiated only after verifying that contralateral visual evoked firing on presumptive LGN electrodes preceded that of V1 electrodes (three animals were not recorded for this reason). For P5-7 animals (prior to visual responsiveness), recordings were initiated if both presumptive LGN or V1 locations displayed prominent 1–5 s firing periods that were interrupted by 10–60 s periods of network silence, indicative of retinal waves at this age (*Hanganu et al., 2006*). Post-mortem reconstruction of the electrode tract was used to verify placement within central dLGN or monocular V1 (*Figure 1B*) by reference to a developmental atlas (*Khazipov et al., 2015*). In total, three P5-7 animals were excluded post-mortem because the electrode was not located in the dLGN. An Ag/AgCl wire was placed over right frontal cortex (~1 mm posterior and 3 mm lateral to *bregma*) as ground. Electrical signals were digitized using the Neuralynx (Bozeman, MT) Digital Lynx S hardware with Cheetah (v5.5) software. dEEG signals were band-pass filtered between 0.1 Hz to 9 kHz, and digitized at 32 kHz. Cortical recordings were referenced to a contact site located in underlying white matter. LGN recordings were referenced to a contact just dorsal to LGN. MUA was extracted by threshold crossing of −50 µV following 300 Hz–9 kHz band-pass. Visual stimuli were provided by a 100 ms whole-field flash (100 lux) every 30 s to the contralateral or ipsilateral eye on a background of low light (<1 lux).

For LGN silencing, the LGN electrode was removed after control recording, and 1 μl of 1 mM Muscimol and 1% Chicago Sky Blue in saline was injected using a Nanoject II (Drummond) at the same stereotaxic coordinates used for the LGN recording. Recordings began 10 min after application of the solution. The injection site was confirmed to be located in LGN in all animals by post-recording histology. Retinal silencing was also performed acutely after obtaining baseline visual responses. The surface of the eye was prepared with 2.5% lidocaine. Eyes were stabilized with blunt forceps and 2–5 μl of APV (5 mM)/CNQX (2 mM) in saline was injected intravitreally under the microscope. Recordings resumed at least 15 min after recovery from anesthesia. Retinal silencing was verified by loss of visual response (n = 2 removed for remaining visual response). For VC silencing, 1–3 μl of APV (5 mM)/CNQX (2 mM) in saline was applied over the small hole above the visual cortex. Recordings started 10 min after application of the solution. VC silencing was confirmed by verifying the complete secession of firing in all layers of VC for the electrodes under the drug placement. Because LGN and VC recordings were not aligned, we cannot be certain that the aligned portion of VC was completely inhibited by the treatment. However, the drug eliminated activity at all depths (~1.2 mm) of VC. Assuming a similar horizontal spread (2 mm diameter), inhibition will, at a minimum, affect all of monocular VC. Recording began 10 min after application and effects were stable for at least 30 min after application, when recording was terminated.

AAV9-Synapsin-promoter-Chronos-GFP (tier 3.5e13 GC/ml) was obtained from Penn Vector Core. P0-1 rat pups were cold anesthetized and 50–100 nl of viral solution was injected into the visual cortex at multiple depths (Nanoject II). At least 5 days post injection yielded strong expression of Chronos-GFP in VC and in cortical axons in LGN, and expression was verified in all animals recorded. To activate Chronos in vivo, a 400 μm diameter optic fiber (Thorlabs, Newton, NJ) coupled with 470 nm LED (Thorlabs) was placed over VC, and 2.5 mW flashes of 470 nm LED illumination were used for activation.

## Analysis

Neural signals were imported into Matlab (Mathworks, MA). Spike-times and dEEG were down sampled to 1 kHz. Before analysis, animals were eliminated for unstable baseline LFPs (fluctuation larger than the maximum amplitude of visual response) or variable spike activity (> 20% change between the start and end of recording) (n = 5). Cortical L4 was identified in each recording as the channel with the earliest negative deflection and the fastest spike response in the mean visual evoked response as previously described (*Berzhanskaya et al., 2016*). For LGN, contralateral- and ipsilateral- responsive regions were identified by 100 ms flashes to either eye. The channel for analysis in LGN was selected by the earliest spike response to contralateral visual stimuli and used for detailed analysis. For P5-7, because of the lack of visual response at this age, cortical L4 was identified as that with the largest spike response during spindle bursts, and the recording site in LGN with the strongest spike activity during spindle bursts was used for detailed analysis.

For all analyses, activity for each animal was calculated from the entirety of a continuous 20 min period prior and 20 min period after manipulation (following 15 min recovery from anesthesia). For spectral analysis in VC, LFP spectra were obtained over a fixed window (2 s for spontaneous and 0.3 s for visually evoked response) by the multitaper method using the freely available Chronux package (*Mitra and Bokil, 2007*) with taper parameters [3 5]. Animal mean spectra were calculated by averaging spectral windows. To reduce the effect of the 1/f relationship, mean multi-taper spectra were multiplied by frequency. For LGN, animal mean MUA spectra were obtained by calculating the fourier transform of the auto-correlation of total MUA. Time-series spectra for LGN MUA used for display (representative traces in *Figures 1*, *3*, *5*) were calculated by filtering the MUA time series with a Gaussian window (5 ms alpha) and applying the multi-taper method as above. For both signals, the frequency axis was resampled on a log scale to equalize the representation of high and low frequencies and reduce the multiple comparisons problem. For normalization, frequency power at each band was divided by the mean 2–55 Hz power.

Events were defined as at least two spikes occurring with intervals of less than 500 ms. Spike continuity was calculated as the proportion of total recording time that contained an event.

## Anatomy

Brains were perfused with 4% paraformaldehyde in PBS, sectioned by vibratome, and mounted with Fluoromount-G (Electron Microscopy Sciences). Images were acquired using a Zeiss 710 confocal microscope with a 10x or 60x objective, or Zeiss DM6000 B microscope with a 10x objective.

## Statistics

All hypothesis testing was conducted using non-parametric because n was less than ten. Wilcoxon signed-rank test was used for pairwise comparison, and mean ± standard error of the mean (SEM) and each data point is reported (*Figures 2A3,B3,C3*, *3C2,C4,D2,D4,E2,E4*, *4G,H* and *5F*). The Kruskal–Wallis test was used for non-parametric comparison of three independent groups, and mean ± SEM and each data point is reported (*Figure 4E and F*). Kolmogorov–Smirnov test was used for non-parametric comparison of distribution of events, and data and 95% confident interval is reported (*Figures 2A4,B4,C4*, *3C5,D5 and E5*). Spectra were examined at each frequency for significant difference using non-parametric permutation tests corrected for multiple comparisons by the method of Cohen (*Cohen, 2014*) (*Figures 2A5,B5,C5*, *3C3,D3,E3* and *5G*). All tests were performed in Matlab. The number of animals is reported in all figures. Exact P-values are presented in all figures; p<0.001 are rounded to nearest power of ten.

# Acknowledgements

We thank Marnie Phillips for extensive assistance in writing and editing the manuscript and guidance on the retinal blockade experiments, and Jason Triplett for comments.

# Additional information

### Funding

| Funder | Grant reference number | Author |
| --- | --- | --- |
| National Eye Institute | EY022730 | Matthew T Colonnese |

The funders had no role in study design, data collection and interpretation, or the decision to submit the work for publication.

### Author contributions

YM, Designed the experiments and wrote the paper, Performed and analyzed the experiments; MTC, Designed the experiments and wrote the paper, Analysis and interpretation of data

### Author ORCIDs

Yasunobu Murata, http://orcid.org/0000-0001-5757-2497
Matthew T Colonnese, http://orcid.org/0000-0002-2480-1270

### Ethics

Animal experimentation: All experiments were conducted with approval from The George Washington University School of Medicine and Health Sciences Institutional Animal Care and Use Committee (Assurance #A-3205-01 and protocol#A245), in accordance with the Guide for the Care and Use of Laboratory Animals (8th Edition, National Academies Press). Surgical proceedures were performed under isolflurane anesthesia, and post-operative pain managed with carprophen and topical anesthetics. All efforts were made to minimize suffering.

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
