## [Decision Letter]

Thank you for submitting your article "An excitatory cortical feedback loop gates retinal wave transmission in thalamus" for consideration by *eLife*. Your article has been reviewed by three peer reviewers, including John Huguenard (Reviewer #2), Jose-Manuel Alonso (Reviewer #3), and Sacha B Nelson (Reviewer #1), who is a member of our Board of Reviewing Editors, and the evaluation has been overseen by David Van Essen as the Senior Editor.

The reviewers have discussed the reviews with one another and the Reviewing Editor has drafted this decision to help you prepare a revised submission.

Summary:

This report follows on this group's highly innovative work documenting novel forms of network responses in the developing brain. The results help explain a key puzzle surrounding the role of spontaneous activity in developing sensory systems. Despite sparse and weak synapses and reduced excitability, peripheral waves are robustly transmitted to central structures. This is an excellent paper that provides very exciting results about the function and development of the corticogeniculate pathway

Essential revisions:

1) It is *eLife* policy to include some reference to the biological system in the title. This often means including reference to the preparation or experimental organism in the title. This might be realized by including the word "mammalian" or "rodent" prior to the word thalamus in the current title.

2) This is the first paper the Reviewing editor has handled for *eLife* that received three reviews with no major points raised. The full reviews are included for your information, but the suggestions for revision are advisory. You may consider this paper acceptable pending minor revision.

*Reviewer #1:*

Murata and Colonnese use simultaneous multi-electrode recording from LGN and visual cortex in awake rats and optogenetic and pharmacological manipulation of retinal, thalamic and cortical activity to demonstrate that corticothalamic feedback amplifies the propagation of retinal waves through the LGN to the cortex. The effect is developmentally regulated and changes to net inhibition around the time of eye opening.

The results help explain a key puzzle surrounding the role of spontaneous activity in developing sensory systems. Despite sparse and weak synapses and reduced excitability, peripheral waves are robustly transmitted to central structures.

The experiments appear to have been carefully done and are very clearly reported. I have no major recommendations for improvement.

*Reviewer #2:*

This report follows on this group's highly innovative work documenting novel forms of network responses in the developing brain. In particular, during the early postnatal period (first two weeks after birth) in rodents there is a distinct mode of thalamocortical network activation with sensory spindles occurring during spontaneous activity (retinal waves) in sensory organs. In this report the authors use silicon probe recordings from dLGN and VC in unanesthetized animal, and make several key observations: 1) early cortical spindles are dependent on thalamic relay of sensory information, 2) corticothalamic feedback produces a very powerful amplification of the sensory signal arising from spontaneous retinal waves, 3) this reinforcement of thalamic activity by cortical feedback during early development disappears later when retinal waves no longer occur, 4) the loss of reinforcement coincided with the development of feedforward inhibition, and 5) the latter appears to be due to cortical recruitment of thalamic inhibition, as inhibitory responses in LGN can be produced by optogenetic activation of deep layer VC neurons.

Overall, the results are interesting, the points well illustrated, and the statistical analysis compelling. I only have a few minor concerns, as follows:

1) In my opinion, there is a dearth of primary data in many of the illustrations. For example, Figure 2 is composed exclusively of derived data, and Figure 3 only shows anecdotal responses in a, and even those are derived spike rasters or spectrograms.

2) Results section, subsection: “Transmission of retinal waves requires corticothalamic feedback”. A more precise location than "just below LGN" would be appropriate.

*Reviewer #3:*

This is an excellent paper that provides very exciting results about the function and development of the corticogeniculate pathway. The experiments are well designed, the data are of high quality and the results are really exciting and beautifully presented. The paper is a pleasure to read and reports several important new results that are likely to be very influential in the field. The main finding that the corticothalamic feedback has a net excitatory effect on LGN at early developmental states and only switches to inhibitory at the end of the postnatal second week (around eye opening) is very interesting and supported by strong and convincing data. Multiple other findings reported in this paper are also interesting and novel. To choose just one example, the finding that silencing the visual cortex reduces LGN activity just as much as silencing of the retina in the Discussion is very interesting and important.

I have no major criticisms. This is one of few rare papers where I find myself making multiple notes in the manuscript not because of problems with the paper but because of novel insights that the authors report and that I want to remember. The significance of the findings is very high and the results have potential important implications both in basic and clinical research. Excellent work!

---

## [Author Response]

Reviewer #2:

*1) In my opinion, there is a dearth of primary data in many of the illustrations. For example, Figure 2 is composed exclusively of derived data, and Figure 3 only shows anecdotal responses in a, and even those are derived spike rasters or spectrograms.*

We revised the first three figures to show example raw data. In Figure 1 we now show the LFP (1-150Hz) and raw MUA (>300+) components of the raw data for both LGN and VC. In Figure 2 and Figure 3 we show representative traces for the relevant (LFP for VC and MUA for LGN) parameter. This should give the reader a more intuitive understanding of the activity in each region and the changes caused by each of the manipulations.

*2) Results section, subsection: “Transmission of retinal waves requires corticothalamic feedback”. A more precise location than "just below LGN" would be appropriate.*

The following replaced “just below LGN”: “…all contacts from the same shank located 20-100µm below the last visually responsive contact…”.